# Health conditions of potential risk for severe Covid-19 in institutionalized elderly people

**Mayara Priscilla Dantas Araújo**[1☺]*, **Vilani Medeiros de Araújo Nunes**[1☺], **Larissa de Albuquerque Costa**[2☺], **Talita Araujo de Souza**[3☺], **Gilson de Vasconcelos Torres**[4☺], **Thaiza Teixeira Xavier Nobre**[3☺]

**1** Department of Collective Health, Federal University of Rio Grande do Norte, Natal, Rio Grande do Norte, Brazil, **2** Municipal Secretary of Health, City Hall of Natal, Natal, Rio Grande do Norte, Brazil, **3** Faculty of Health Sciences of Traíri, Federal University of Rio Grande do Norte, Santa Cruz, Rio Grande do Norte, Brazil, **4** Department of Nursing, Federal University of Rio Grande do Norte, Natal, Rio Grande do Norte, Brazil

☺ These authors contributed equally to this work.
* mayaraaraujonutri@gmail.com

**Data Availability Statement:** All relevant data are within the manuscript and its Supporting information files.

## Abstract

The objective of this study was to identify the health conditions considered potential risk factors for severe Covid-19 and analyze its association with the BMI of elderly people living in Long-Term Care Facilities (LTCF). This is a descriptive and cross-sectional study, with a quantitative approach, carried out in eight LTCF in the Metropolitan Region of Natal, Rio Grande do Norte, with a population of 267 elderly people, between the months of February and December 2018. The Elderly Health Handbook was used to collect data on sociodemographic, health and risk factors. The Pearson's Chi-square test and odds ratio were used for the analysis. A higher frequency of low weight was observed in elderly people with cognitive impairment (24.6%), and overweight in those hypertensive (23.3%) and diabetics (12.9%). BMI was associated with the age group of 80 years or over, hypertension and diabetes (p = 0.013; p < 0.001; p = 0.001). Hypertensive elderly people were more likely to have low weight when compared to non-hypertensive individuals (RC = 3.6; 95% CI 1.5–8.6). The institutionalized elderly individuals present health conditions that may contribute to the occurrence of adverse outcomes in case of infection by Covid-19. The importance of protective measures for this population must be reinforced, in view of the devastating action of this disease in these institutions.

## Introduction

Seven types of human coronavirus are known, including SARS-CoV-1 which causes severe acute respiratory syndrome, and the most recent that has been named SARS-CoV-2, the virus that causes Covid-19 disease. This disease can cause infections with symptoms initially similar to a cold or mild flu, but with the risk of worsening to the point of death [1, 2].

The United States Center for Disease Control and Prevention [3] highlights the fact of living in Long-Term Care Facilities for the Elderly (LTCF), more severe chronic heart and lung

**Funding:** The project was funded by the International Network for Research on Vulnerability, Health, Safety and Quality of Life of the Elderly: Brazil, Portugal and Spain, by the Edital 01/2020 of Pro-Rectory of Research of Federal University of Rio Grande do Norte (PROPESQ - UFRN). GVT is a productivity fellow by the National Council for Scientific and Technological Development (CNPq) [process 309213/2017-7] and MPDA is a fellow by the Coordenação de Aperfeiçoamento de Pessoal de Nível Superior - Brazil (CAPES) - Finance Code 001.

**Competing interests:** We have no conflicts of interest to disclose.

diseases, chronic kidney disease on dialysis, liver disease, immunosuppressed condition and severe obesity (Body Mass Index–BMI $> 40$ kg/m$^2$) as high-risk factors for developing the severe form of Covid-19.

Due to advanced age and because they are subjected to multi-morbidities, elderly people are more vulnerable to evolving to a severe form of Covid-19, especially those who live in LTCF, since the institutionalization itself leads to frailty, which, in turn, is associated with functional, cognitive and psychological deficits and ultimate loss of autonomy and independence in some elderly patients [4].

The consequences of Covid-19 in LTCF have been devastating. Countries such as the United States, Spain and Italy have had a high mortality rate among institutionalized elderly people. The proximity among residents leads to a higher risk of infection and, consequently, adverse outcomes, with mortality associated with difficulty in preventing the spread of the virus in the institutions [5].

Thus, it is necessary to adopt care measures to protect elderly people and prevent adverse health events resulting from Covid-19. Among these precautions, one can mention the monitoring and maintenance of an adequate nutritional status, since malnutrition and overweight/obesity lead to changes in the immune response, thus contributing to greater susceptibility to the new coronavirus [6].

Motivated by this need, it was sought to know the health conditions of elderly people living in LTCF with regard to their nutritional status and comorbidities, in order to identify the existence of factors that may constitute a risk for worse evolution in case of Covid-19 infection.

In view of the situation of vulnerability faced by elderly people and based on the assumption that nutritional status is associated with risk factors among this population, this study aimed to identify the health conditions considered potential risk factors for severe Covid-19 and analyze its association with the BMI of institutionalized elderly citizens.

## Materials and methods

Descriptive, cross-sectional study with a quantitative approach carried out in eight LTCF in the metropolitan region of Natal, Rio Grande do Norte (RN), six in the municipality of Natal, one in Parnamirim and one in Macaíba, which are surrounding towns.

The study had as participants all the elderly residents of the eight LTCF registered in the city of Natal and in the Metropolitan region, RN, and who were present at the institution at the time of data collection, totaling 267 people.

Data collection took place between February and December 2018 as part of the study entitled "Monitoring the Health of the Elderly" at the Federal University of Rio Grande do Norte.

The intention of the study to seek information in LTCF emerged as a consequence of the advent of the COVID-19 pandemic in December 2019 and considering that the group with the greatest vulnerability is the elderly population. In these institutions, a relatively significant number of elderly people live in groups. Accordingly, the study aimed to identify situations of vulnerability based on information regarding health conditions and nutritional aspects during the collection period, and potential risk factors for severe Covid-19, also present in the evaluation of all the individuals included in the study.

In order to obtain data, information from the Elderly Health Handbook (EHH), version 2017, prepared by the Brazilian Ministry of Health, was used as a collection instrument, supplemented with information from the records of the elderly citizens in the institutions where they lived. EHH allows the identification of individual health needs and potential for risk and degrees of frailty, in addition to making it possible to make a comprehensive health assessment and identify the main vulnerabilities [7].

The data collected includes the sociodemographic profile, such as gender (female and male), age (classified into age groups: 60–79 and $\geq 80$), schooling (illiterate, 1–3, 4–7 and $\geq 8$ years of study), race (white and non-white) and marital status (single, widowed and divorced/separated), nutritional status, according to the Body Mass Index (BMI) classification, current life habits (tobacco and alcohol consumption), and health conditions that presented medical diagnosis, among them are hypertension, diabetes, cognitive impairment, coronary disease, asthma, and chronic obstructive pulmonary disease (COPD).

The outcome variable of this study was BMI, obtained by dividing weight in kilograms (kg) by height in meters squared ($m^2$) and classified according to EHH [7], low weight is indicated by BMI < 22 kg/$m^2$, normal weight by BMI $\geq$ 22 kg/$m^2$ and <27 kg/$m^2$, and over-weight by BMI $\geq$ 27 kg/$m^2$. The independent variables were the health conditions and smoking which, according to the pertinent literature, are risk factors for severe Covid-19 in the elderly person [8, 9].

Data were tabulated and organized in the Excel® software, version 2010 (Microsoft Office), and the statistical, descriptive and inferential analysis was performed in the Statistical Package for the Social Sciences (SPSS), version 21.0. In order to assess the association between nutritional status and risk factors presented by the elderly person, bivariate analyses were performed with a significance level of 5% ($p \leq 0.05$), using Pearson's chi-square test and odds ratio (OR) with a 95% Confidence Interval (95% CI).

The study was approved by the Research Ethics Committee of the Onofre Lopes Hospital (CEP/HUOL) under the opinion number 2.366.555 and CAAE: 78891717.7.0000.5292.

## Results

Of the 267 elderly people evaluated, 32 (12.0%) presented some missing data. Of all the participants, only data regarding gender and age were available. For this reason, there were variations in the total number of participants for the variables under analysis.

There was a higher frequency of women, aged 80 years or over, with an average age of 80.3 ± 9.1 years, single, with no education, and non-white race/color. As for life habits, there was a greater distribution of non-smokers and non-consumers of alcohol. Regarding nutritional status, according to BMI, elderly people with low weight predominated, with an average value of 24.8 ± 15.2 kg/$m^2$ (Table 1).

The most frequent risk conditions for severe COVID-19 were cognitive impairment (135, 57.5%), hypertension (130, 55.6%), diabetes (66, 27.5%), coronary disease (15, 6.3%), asthma (7, 2.9%) and COPD (5, 2.1%). All the diseases evaluated were more prevalent in females when compared to males.

When associating nutritional status with risk factors for severe Covid-19, there was a statistically significant difference between the age group of 80 years or over and low weight ($p = 0.013$), between hypertension and overweight ($p < 0.001$) and between absence of diabetes and all the BMI categories ($p = 0.001$) (Table 2).

It was identified that elderly people aged 80 years or over with or without hypertension were mostly low weight, while elderly people aged 60 to 79 years with hypertension were mostly overweight, and those without hypertension had mostly adequate weight. A significant association was found between the age group of 60 to 79 years without hypertension and adequate BMI, and between this age group with hypertension and overweight ($p < 0.001$), as shown in Table 3.

When assessing the association between hypertension and low weight, it was observed that hypertensive elderly people aged 80 years or over had a greater tendency towards low weight when compared to elderly people between 60 and 79 years old. It was found that, among

**Table 1. Description of the characteristics of institutionalized elderly people in the municipality of Natal and metropolitan region, Rio Grande do Norte, Brazil, 2020.**

| Sociodemographic characterization [a] | | n | % |
|---|---|---|---|
| Sex | Female | 185 | 69.3 |
| | Male | 82 | 30.7 |
| Age group in years | 60 to 79 | 122 | 45.7 |
| | ≥ 80 | 145 | 54.3 |
| Marital status | Not married | 124 | 49.2 |
| | Widowed | 75 | 29.8 |
| | Divorced/separated | 53 | 21.0 |
| Schooling in years | Illiterate | 92 | 39.0 |
| | 1 to 3 | 55 | 23.3 |
| | 4 to 7 | 47 | 19.9 |
| | 8 or more | 42 | 17.8 |
| Race | Non-white | 129 | 51.2 |
| | White | 123 | 48.8 |
| Smoker | No | 224 | 86.5 |
| | Yes | 35 | 13.5 |
| Alcohol consumption | No | 244 | 94.2 |
| | Yes | 15 | 5.8 |
| BMI | Low weight | 91 | 37.6 |
| | Eutrophy | 86 | 35.5 |
| | Overweight | 65 | 26.9 |

BMI, Body Mass Index.

[a] Variable total number of respondents due to the lack of information.

hypertensive elderly people, the chance of being underweight was approximately 4 times greater than that of elderly people who did not have hypertension (OR = 3.6; 95% CI 1.5–8.6). There was statistical significance between the age group of 80 years or over and low weight (Table 4).

In the association of hypertensive elderly people by age group with the presence of overweight, the same results were identified for the two age groups. Both obtained a frequency of 20.6% overweight, with no significant difference between the variables (Table 5).

## Discussion

In the present study, the findings referring to sociodemographic characteristics corroborate with the literature, which shows a higher frequency of elderly women, aged 80 years or over, single, with or without schooling, and non-white people living in LTCF [10, 12].

The predominance of women is a reflection of the distribution of this gender in the world population and their higher life expectancy. This, associated with the lack of a partner and family support for care, make elderly people, especially women, more vulnerable to institutionalization [4, 11].

The low level of education frequently observed in these people may be the result of the difficulty of access and devaluation of formal education [11], while the predominance of the non-white race may be related to the institution, as observed by Pinheiro et al. [11], who identified an association between race with the type of institution, with a predominance of non-white elderly citizens in non-white in non-profit LTCF and white people in for-profit institutions.

**Table 2. Association between risk factors for severe Covid-19 and BMI of institutionalized elderly people in the municipality of Natal and metropolitan region, Rio Grande do Norte, Brazil, 2020.**

| Risk factors for severe Covid-19 | | BMI | | | | | | P-value [a] |
|---|---|---|---|---|---|---|---|---|
| | | Low weight | | Eutrophy | | Overweight | | |
| | | n | % | n | n | % | n | |
| Age Group [b] | 60 to 79 | 30 | 12.5 | 42 | 17.5 | 36 | 15.0 | 0.013 |
| | ≥ 80 | 61 | 25.4 | 36 | 15.0 | 35 | 14.6 | |
| Cognitive disability | Yes | 59 | 24.6 | 44 | 18.3 | 37 | 15.4 | 0.243 |
| | No | 32 | 13.3 | 34 | 14.2 | 34 | 14.2 | |
| Hypertension | Yes | 38 | 15.8 | 42 | 17.5 | 56 | 23.3 | <0.001 |
| | No | 53 | 22.1 | 36 | 15.0 | 15 | 6.3 | |
| Diabetes | Yes | 15 | 6.3 | 20 | 8.3 | 31 | 12.9 | 0.001 |
| | No | 76 | 31.7 | 58 | 24.2 | 40 | 16.7 | |
| Smoking | Yes | 11 | 4.6 | 16 | 6.7 | 7 | 2.9 | 0.136 |
| | No | 80 | 33.3 | 62 | 25.8 | 64 | 26.7 | |
| Coronary disease | Yes | 6 | 2.5 | 5 | 2.1 | 4 | 1.7 | 0.967 |
| | No | 85 | 35.4 | 73 | 30.4 | 67 | 27.9 | |
| Asthma | Yes | 4 | 1.7 | 3 | 1.3 | 0 | 0.0 | 0.215 |
| | No | 87 | 36.3 | 75 | 31.3 | 71 | 29.6 | |
| COPD | Yes | 1 | 0.4 | 1 | 0.4 | 3 | 1.3 | 0.321 |
| | No | 90 | 37.5 | 77 | 32.1 | 68 | 28.3 | |

BMI, Body Mass Index; COPD, Chronic Obstructive Pulmonary Disease.

[a] Pearson's chi-square test.

[b] Variable total number of respondents due to the lack of information.

These characteristics can contribute to the presence of cognitive impairment in these people, the most prevalent condition among the elderly people analyzed, resulting in impaired quality of life and health conditions [10]. Furthermore, institutionalized elderly people have worse health conditions due to the presence of multi-comorbidities, leading to the greatest frailty [4, 12].

Regarding the age group, the predominance of elderly people aged 80 years or over found in this study may be a consequence of the forms of dependence that come with aging, making people more susceptible to chronic and weakening diseases, thus contributing to their institutionalization [4]. For this reason, due to the decline in immune function and chronic low-

**Table 3. Association of age group and hypertension with the nutritional status of elderly people living at the LTCF of the municipality of Natal and metropolitan region, Rio Grande do Norte, Brazil, 2020.**

| Associated risk factors for severe Covid-19 | BMI | | | | | | Total | | P-value [a] |
|---|---|---|---|---|---|---|---|---|---|
| | Low weight | | Eutrophy | | Overweight | | | | |
| | n | % | n | % | n | % | n | % | |
| 60 to 79 without hypertension | 22 | 9.2 | 36 | 15.0 | 8 | 3.3 | 66 | 27.5 | < 0.001 |
| ≥ 80 without hypertension | 31 | 12.9 | 0 | 0.0 | 7 | 2.9 | 38 | 15.8 | |
| 60 to 79 with hypertension | 8 | 3.3 | 20 | 8.3 | 29 | 12.1 | 57 | 23.8 | |
| ≥ 80 with hypertension | 30 | 12.5 | 22 | 9.2 | 27 | 11.3 | 79 | 32.9 | |
| Total | 91 | 37.9 | 78 | 32.5 | 71 | 29.6 | 240 | 100.0 | |

BMI, Body Mass Index.

[a] Pearson's chi-square test.

**Table 4. Association of hypertension by age group with low weight in institutionalized elderly people in the municipality of Natal and metropolitan region, Rio Grande do Norte, Brazil, 2020.**

| Presence of hypertension by age group | Low weight | | | | Total | | P-value [a] OR (95% CI) |
| --- | --- | --- | --- | --- | --- | --- | --- |
| | Absent | | Present | | | | |
| | n | % | n | % | n | % | |
| 60 to 79 years | 48 | 35.3 | 8 | 5.9 | 56 | 41.2 | 0.003 3.6 (1.5–8.6) |
| ≥ 80 years | 50 | 36.8 | 30 | 22.1 | 80 | 58.8 | |
| Total | 98 | 72.1 | 38 | 27.9 | 136 | 100.0 | |

OR, Odds ratio; CI, 95% confidence interval.

[a] Pearson's chi-square test.

**Table 5. Analysis of the association of hypertension by age group with overweight in institutionalized elderly people in the municipality of Natal and metropolitan region, Rio Grande do Norte, Brazil, 2020.**

| Presence of hypertension by age group | Overweight | | | | Total | | P-value [a], CR (95% CI) |
| --- | --- | --- | --- | --- | --- | --- | --- |
| | Absent | | Present | | | | |
| | n | % | n | % | n | % | |
| 60 to 79 years | 28 | 20.6 | 28 | 20.6 | 56 | 41.2 | 0.080, 0.5 (0.3–1.0) |
| ≥ 80 years | 52 | 38.2 | 28 | 20.6 | 80 | 58.8 | |
| Total | 80 | 58.8 | 56 | 41.2 | 136 | 100.0 | |

OR, Odds ratio; CI, 95% confidence interval.

[a] Pearson's chi-square test.

grade systemic inflammation [13], elderly people are in the group at risk for severe Covid-19 if contaminated [14].

Among the main findings of this study, one can mention the association of the age group with the nutritional status, with low weight predominating over the other categories, especially among elderly people aged 80 years or over. This is consistent with the pertinent literature, which highlights the predominance of elderly people at risk for nutritional status or malnutrition in LTCF, a common condition in this public, especially among the oldest individuals, due to the decline in nutritional status that takes place with advancing age [10, 15].

A study carried out in China [16] found a predominance of 52.7% of malnutrition and 27.5% of risk of malnutrition in elderly patients with Covid-19, with diabetes being one of the factors associated with malnutrition. A similar result was found by Li et al. [17], with malnutrition, hypertension and diabetes being more frequent in patients with the severe form of Covid-19. Although a higher frequency of low weight was found in the elderly individuals evaluated in this study, this condition was more frequent in those who did not have hypertension and diabetes when compared to those who presented these comorbidities.

Elderly people affected by Covid-19 may be more susceptible to malnutrition [16, 18]. This may be due to the presence of gastrointestinal symptoms, presented by most infected elderly people [8, 10], which directly interfere with food consumption and nutritional status, or by the deleterious effects of the acute inflammatory response to SARS-CoV-2 [16]. A worrying factor is that gastrointestinal symptoms are frequent in patients with rapid progression and worse disease outcome [19].

Just like malnutrition, overweight causes damage to the immune system, increasing the frequency and severity of infectious diseases, thus leading to a greater risk of hospitalization,

serious diseases and mortality [15, 20]. Despite this, only obesity is identified as a risk factor for the severe form of Covid-19 [21].

The results of this study show a lower frequency of overweight among institutionalized elderly residents; however, this is a condition that requires care since studies show obesity as an important predictor of hospitalization and worsening of Covid-19 infection, with high BMI values associated with greater chances of hospitalization [22] and need for invasive mechanical ventilation [21].

Obesity also contributes to the emergence of chronic diseases such as diabetes and hypertension [20], thus reinforcing the findings of the present study. Here, a higher frequency of overweight in elderly people who presented such comorbidities was found. Such findings have been associated with worst outcomes of Covid-19 due to the high prevalence of hypertension and diabetes among elderly patients who died due to this disease [14, 23], which suggests that such comorbidities are important risk factors for worsening and mortality due to Covid-19.

However, although overweight is associated with the presence of these comorbidities, when the presence of hypertension was associated with the age group, it was found that hypertensive elderly people were more likely to present low BMI when compared to non-hypertensive people. Low weight was more frequent in hypertensive elderly people aged 80 years or over. This finding differs from that observed in the pertinent literature, which does not point to hypertension as a risk factor for low weight [24], but rather for diabetes [16].

In addition to advanced age being a risk factor and the mortality rate increasing according to age [13], a study showed that hypertensive elderly people have worse outcomes for Covid-19 [8], and patients with severe conditions with hyperglycemia had a higher risk of death [25]. This emphasizes the need for continuity of care and control of comorbidities, in order to prevent the disease from evolving to a severe form if elderly people are infected.

The higher frequency of comorbidities can make the elderly more susceptible to worse Covid-19 outcomes. Among these comorbidities, the most prevalent are hypertension, diabetes, cardiovascular and respiratory diseases [26]. A similar result was observed in this study where, among the comorbidities presented by the institutionalized elderly people, the most frequent was cognitive impairment, followed by hypertension, diabetes, cardiovascular diseases and respiratory diseases. McMichael et al. [27] evaluated the existing comorbidities in 101 elderly people institutionalized with Covid-19 and identified hypertension as the most frequent, followed by heart disease, kidney disease, diabetes and obesity.

It was observed that the elderly people evaluated had many of the factors considered potential risk for adverse outcomes, which made them more vulnerable to severe Covid-19 in case of infection, both because of their health conditions and because they lived in LTCF, which is one of the environments conducive to the worst outcomes for the new coronavirus [27].

Thus, maintaining good nutritional status can contribute to reducing complications from comorbidities and Covid-19. Infected, at-risk or recovered patients need adequate nutritional support because viral infections are associated with nutritional deficiencies, and healthy eating is important for all elderly people [28].

The main limitation of this study was the use of BMI only to assess nutritional status; although studies on Covid-19 use this index, the measurement of other anthropometric indicators and the assessment of biochemical and dietetic parameters, for example, could contribute to a more accurate nutritional diagnosis. With this, this study can become a reference for further research, where more robust nutritional assessments in institutionalized elderly people who have been affected by Covid-19, observing how the disease interferes with the nutritional status of these individuals.

Although data were collected in 2018, before the occurrence of the Covid-19 pandemic, a maintenance of the epidemiological profile of institutionalized elderly people is still observed,

which is similar to the epidemiological profile of the non-institutionalized elderly population, with an increase in the presence of chronic diseases with age and, especially among females [29]. Thus, the results brought in this study reflect the health conditions of these elderly people and can assist in health actions aimed to minimize the risks and adverse outcomes of Covid-19 in this audience.

## Conclusions

The results showed that institutionalized elderly people have chronic morbidities considered to have potential risk for adverse outcomes due to SARS-CoV-2 infection. When associated with the deficit in nutritional status, the advanced age and the presence of chronic conditions may contribute to increasing the vulnerability of these individuals to severe Covid-19, especially the presence of hypertension, one of the main risk factors described in the pertinent literature, which leads to greater chances of causing underweight elderly individuals when compared to those without hypertension.

The findings of this study underline the importance of protective measures against Covid-19 in LTCF, in view of the devastating action of this disease in these institutions. It is essential to define strategies for the care of this population, in order to improve health and nutritional conditions.

## Supporting information

**S1 File.**
(DOCX)

**S1 Checklist.**
(DOC)

## Author Contributions

**Conceptualization:** Mayara Priscilla Dantas Araújo, Vilani Medeiros de Araújo Nunes, Larissa de Albuquerque Costa, Talita Araujo de Souza, Gilson de Vasconcelos Torres, Thaiza Teixeira Xavier Nobre.

**Data curation:** Mayara Priscilla Dantas Araújo, Vilani Medeiros de Araújo Nunes, Larissa de Albuquerque Costa, Talita Araujo de Souza, Gilson de Vasconcelos Torres, Thaiza Teixeira Xavier Nobre.

**Formal analysis:** Mayara Priscilla Dantas Araújo, Vilani Medeiros de Araújo Nunes, Larissa de Albuquerque Costa, Talita Araujo de Souza, Gilson de Vasconcelos Torres, Thaiza Teixeira Xavier Nobre.

**Investigation:** Mayara Priscilla Dantas Araújo, Vilani Medeiros de Araújo Nunes, Larissa de Albuquerque Costa, Talita Araujo de Souza, Gilson de Vasconcelos Torres, Thaiza Teixeira Xavier Nobre.

**Methodology:** Mayara Priscilla Dantas Araújo, Vilani Medeiros de Araújo Nunes, Larissa de Albuquerque Costa, Talita Araujo de Souza, Gilson de Vasconcelos Torres, Thaiza Teixeira Xavier Nobre.

**Resources:** Mayara Priscilla Dantas Araújo, Vilani Medeiros de Araújo Nunes, Larissa de Albuquerque Costa, Talita Araujo de Souza, Gilson de Vasconcelos Torres, Thaiza Teixeira Xavier Nobre.

**Visualization:** Mayara Priscilla Dantas Araújo, Vilani Medeiros de Araújo Nunes, Larissa de Albuquerque Costa, Talita Araujo de Souza, Gilson de Vasconcelos Torres, Thaiza Teixeira Xavier Nobre.

**Writing – original draft:** Mayara Priscilla Dantas Araújo, Vilani Medeiros de Araújo Nunes, Larissa de Albuquerque Costa, Talita Araujo de Souza, Gilson de Vasconcelos Torres, Thaiza Teixeira Xavier Nobre.

**Writing – review & editing:** Mayara Priscilla Dantas Araújo, Vilani Medeiros de Araújo Nunes, Larissa de Albuquerque Costa, Talita Araujo de Souza, Gilson de Vasconcelos Torres, Thaiza Teixeira Xavier Nobre.

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
