## [Decision Letter · Decision Letter 0]

16 Nov 2020

PONE-D-20-33451

Association between nutritional status and risk factors for COVID-19 in institutionalized elderly people

PLOS ONE

Dear Dr. Araújo,

Thank you for submitting your manuscript to PLOS ONE. After careful consideration, we feel that it has merit but does not fully meet PLOS ONE’s publication criteria as it currently stands. Therefore, we invite you to submit a revised version of the manuscript that addresses the points raised during the review process.

We look forward to receiving your revised manuscript.

Kind regards,

Rasheed Ahmad, Ph.D.

Academic Editor

PLOS ONE

Journal Requirements:

3. In your statistical analyses, please state whether you accounted for clustering by region. For example, did you consider using multilevel models?

4. Please provide further details on sample size and power calculations.

5. As part of your revision, please complete and submit a copy of the STROBE checklist, a document that aims to improve reporting and reproducibility of observational studies for purposes of post-publication data analysis and reproducibility: (http://www.strobe-statement.org).

Please include your completed checklist as a Supporting Information file. Note that if your paper is accepted for publication, this checklist will be published as part of your article.

6. Thank you for stating the following financial disclosure: 'NO'

7. Thank you for stating the following in your Competing Interests section: 'NO'

a. Please complete your Competing Interests statement to state any Competing Interests. If you have no competing interests, please state "The authors have declared that no competing interests exist.", as detailed online in our guide for authors at http://journals.plos.org/plosone/s/submit-now

Reviewers' comments:

Reviewer's Responses to Questions

**Comments to the Author**

1. Is the manuscript technically sound, and do the data support the conclusions?

Reviewer #1: No

2. Has the statistical analysis been performed appropriately and rigorously? 

Reviewer #1: Yes

3. Have the authors made all data underlying the findings in their manuscript fully available?

Reviewer #1: Yes

4. Is the manuscript presented in an intelligible fashion and written in standard English?

Reviewer #1: Yes

5. Review Comments to the Author

Reviewer #1: This manuscript included the epidemiological data of a specific population in the capital of a Brazilian state. For the analysis, individuals were selected from a period when the COVID-19 pandemic did not exist. Individual characteristics may have changed over the period between data collection and the outbreak of the epidemic. There is a gap of approximately one year between data collection and the start of the pandemic. In addition, it was not clear why the data found were considered risk factors for severe forms of COVD-19 infection, since there are no reports that the study subjects developed the disease. I do not consider the data in the present study to be relevant, since a direct relationship between risk factors and the development of the disease in its various forms has not been demonstrated.

6. PLOS authors have the option to publish the peer review history of their article (what does this mean?). If published, this will include your full peer review and any attached files.

Reviewer #1: No

---

## [Author Response · Author response to Decision Letter 0]

2 Jan 2021

We identified errors that compromised the understanding of our study. We made the changes in the manuscript in order to improve its understanding. With this, we realized the need to rewrite the objective and the conclusion in order to ensure that our study meets the comments of the reviewer and the journal.

Although the Covid-19 pandemic did not exist when the study data were collected, we observed, based on the literature, that the epidemiological profile of the elderly analyzed resembles that of the elderly domiciled and that of the world population. Moreover, the nutritional profile of the elderly is also maintained, with weight reduction as age progresses. Thus, we understand that the analyzed data can provide support to the planning of protection actions for this public because their health conditions can contribute to the worsening of Covid-19 in case of infection and the deleterious effects of this disease in these institutions already reported worldwide.

We observed that in some parts we did not make ourselves understood correctly. The writing made the reader understand that the presence of risk factors would lead to the development of the disease, instead of these risk factors can lead to the development of the severe form of Covid-19 in case of infection, as established in the literature. In this way, we make the changes throughout the text for a complete understanding of our results and conslusion.

---

## [Editor Report · Decision Letter 1]

4 Jan 2021

Health conditions of potential risk for severe Covid-19 in institutionalized elderly people

PONE-D-20-33451R1

Dear Dr. Araújo,

We’re pleased to inform you that your manuscript has been judged scientifically suitable for publication and will be formally accepted for publication once it meets all outstanding technical requirements.

Kind regards,

Rasheed Ahmad, Ph.D.

Academic Editor

PLOS ONE
---

## [Editor Report · Acceptance letter]

7 Jan 2021

PONE-D-20-33451R1 

Health conditions of potential risk for severe Covid-19 in institutionalized elderly people 

Dear Dr. Araújo:

I'm pleased to inform you that your manuscript has been deemed suitable for publication in PLOS ONE. Congratulations! Your manuscript is now with our production department. 

Kind regards, 

on behalf of

Dr. Rasheed Ahmad 

Academic Editor

PLOS ONE